# CLA-RA: Collaborative Active Learning Amidst Relabeling Ambiguity

## Abstract

Obtaining diverse and high-quality labeled data for training efficient classifiers remains a practical challenge. Crowdsourcing, which involves employing multiple weak labelers, is a popular approach to address this issue. However, crowd labelers often introduce noise, inaccuracies, and possess limited domain knowledge. In this paper, we propose a novel framework CLA-RA to optimize the labeling process by determining what to label next and assigning tasks to the most suitable annotators. Our technique aims to optimize classifier efficiency by utilizing the collective wisdom of various annotators while limiting the influence of error-prone annotations. The key contributions of our work include an annotator disagreement based instance selection mechanism which identifies the noise present in annotations of the instances and an instance-dependent annotator confidence model, which identifies the annotator with the highest confidence to correctly label an instance. These methods, combined with a similarity based annotator inference method, result in improved classifier accuracy while reducing annotation efforts. Experimental results over 13 datasets demonstrate significant improvements over state-of-the-art multi-annotator active learning methods, highlighting the effectiveness of our approach in obtaining high-quality labeled data for training classifiers with minimal labeling costs and errors.

## 1 Introduction

Supervised learning algorithms demand a substantial quantity of labeled data to generate dependable models. Although unlabeled data is abundant and cost-effective to acquire, the process of obtaining class labels entails extensive human involvement. As a result, the development of intelligent learning algorithms to reduce labeling costs have garnered substantial research attention. Active learning has proven to be a valuable approach for training machine learning models by iteratively selecting informative instances for annotation, reducing annotation effort while maintaining or improving classifier performance Cohn et al. (1994); Settles (2009). Traditionally, this process assumes the availability of an omniscient oracle capable of providing correct labels for all queried instances Herde et al. (2021); MacKay (1992). However, in many real-world scenarios, obtaining labeled data solely from a single expert annotator proves challenging, necessitating the utilization of multiple weak annotators, such as crowd workers, to enrich the labeling process and accelerate data annotation process. This new active learning paradigm presents a new challenge as conventional methods, designed for an omniscient oracle, struggle to handle the uncertainties and potential inaccuracies introduced by these weak labelers Urner et al. (2012); Donmez et al. (2009); Rashidi & Cook (2011). Recent research efforts have recognized this issue, leading to the emergence of methods attempting to extract valuable labeling information from multiple imperfect annotators Raykar et al. (2010); Dekel & Shamir (2009); Yan et al. (2011; 2014). However, while leveraging multiple annotators can increase annotation throughput, it introduces new challenges, particularly due to the inherent noise, inaccuracies, and limited domain knowledge exhibited by these crowd-labelers Paun et al. (2018). The presence of such errors can adversely impact the classifier's generalization ability and overall performance.

In this paper, we propose a novel framework, CLA-RA [1], that harnesses the diverse perspectives by utilizing the collective wisdom of various annotators (oracles) while effectively mitigating the

---

[1]CLA-RA stands for **C**ollaborative Active **L**earning **A**midst **R**elabeling **A**mbiguity

impact of annotator errors on classifier performance. The core idea behind CLA-RA is to recognize that different annotators possess varying degrees of proficiency across distinct regions of the feature space. By inferring the expertise level of each annotator and correlating it with the complexity of the instances, our approach optimizes the annotation process by determining which instances to label next and assigning them to the most suitable annotators. Consequently, this strategy yields higher-quality labels and improves classifier effectiveness. Our contributions can be summarized as follows:

- We develop an annotator agreement-disagreement based instance selection approach, which aims to query instances with higher label noise due to disagreement amongst annotators.

- We propose an instance-dependent annotator confidence model that tailors the annotator selection to the unique characteristics of instances, significantly enhancing the precision of annotator assignment.

- We introduce an annotator inference model based on past annotation knowledge which labels certain instances without the need of querying the annotator.

- We conduct extensive experiments on 13 diverse datasets to emperically demostrate the effectiveness of our approach over existing SOTA methods.

## 2 RELATED WORK

Many active learning (AL) strategies assume that there is a single, omniscient annotator who can provide correct labels for the instances. A comprehensive overview of these strategies can be found in Fu et al. (2013). However, with the emergence of Web-based applications has led to active learning problems involving multiple non-expert labelers providing weak labels for the same instances. To address this challenge, researchers have proposed methods that aggregate annotations from multiple annotators. In this section, we will discuss related AL strategies that take into account the possibility of error-prone annotators.

There have been numerous active learning algorithms that consider the availability of not only multiple annotators but also the prospect of them being incorrect. In 2009, IEThresh introduced by Donmez et al. (2009) considered imperfect annotators. Relabeling is a common approach for handling multiple noisy annotators, involving repeated queries and majority voting for label aggregation. In 2010, IEAdjCost introduced by Zheng et al. (2010) extended IEThresh. It assessed annotator proficiency and excluded low-performing annotators. A drawback of the above methods is their unrealistic assumption of uniform performance across instances. Proactive, introduced by Moon & Carbonell (2014), estimates class-dependent annotator performance, using known annotations when available or approximations. Instance and Annotator selection occur simultaneously using uncertainty sampling. Later CEAL Huang et al. (2017) and ALIO by Chakraborty (2020) take a step further by assuming instance-dependent annotation performances. Another research direction emphasizes relabeling samples as a means to reduce noise. Wauthier & Jordan (2011) active learning algorithm can balance relabeling and obtaining labels for new examples. However, they do not explicitly identify or explore this tradeoff, and their solution depends on gold labels and a custom classifier. Lin et al. (2016) requires relabelling each instance multiple times. Very recently DAAL, presented by Baumler et al. (2023) first estimates a model that predicts annotator entropy trained using very few multiply-labeled examples. Later it collects annotations on examples where the entropy of the estimated model and annotator entropy are the most different. Existing approaches exhibit limitations such as they either demand substantial computational resources, rely on initial ground truth labels for model initialization, or presume prior knowledge regarding annotator expertise. In response to these challenges, our paper introduces a novel framework. This framework leverages the collective knowledge of multiple annotators and intelligently assigns instances to the most suitable annotators by carefully balancing the trade-off between exploration and exploitation.

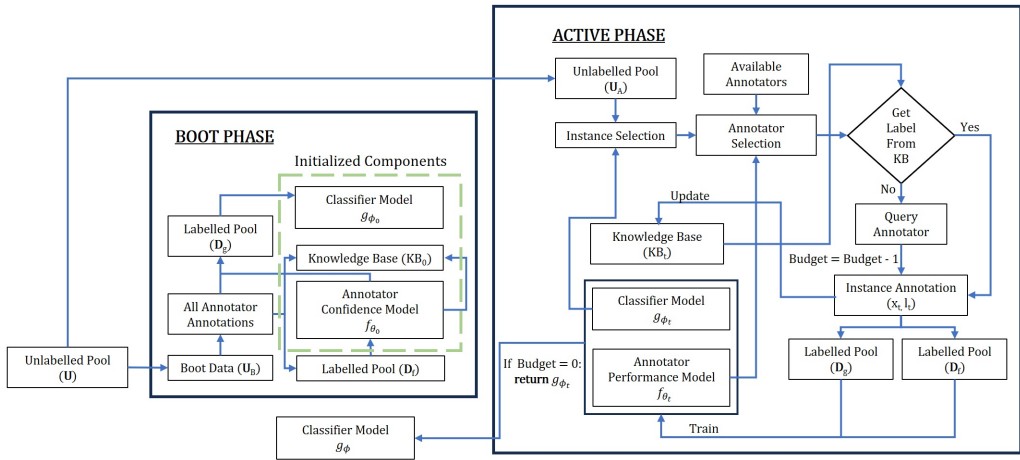

Figure 1: The Active Learning Framework of CLA-RA

# 3 METHODOLOGY

## 3.1 PROBLEM DEFINITION

Let $\mathcal{X}$ be the input feature space and $\mathcal{Y}$ be the label space where $|\mathcal{Y}| = C$ $C$ being the number of classes. Given a set of unlabeled instances $U = \{x_i\}_{i=1}^M$ where $x = (x_1, \ldots, x_k)^T \in \mathcal{X}$, a budget $B$ and a set of annotators $A = \{A_1, \ldots, A_N\}$ where each annotator $A_j$ can be queried for an instance $x_i$ to obtain a label $l_{ij}$, which could be noisy. The goal is to train a classifier model $g_\phi : \mathcal{X} \to \mathcal{Y}$ which predicts the true class label $y$ for a given instance $x$.

To solve the above problem, we propose a novel multi-annotator active learning framework CLA-RA. It consists of an instance selection strategy to select the instances to be queried, an annotator selection strategy that determines which annotator to query for a given instance, and a knowledge base that gets populated during the active learning phase and facilitates a reduction in subsequent queries. We elaborate on each of the components of our framework in detail in the following sections.

## 3.2 CLA-RA FRAMEWORK

Our framework consists of an instance selection strategy (AAD) which combines uncertainty-based selection with annotator agreement-disagreement to determine which instance needs to be queried for a new label. We also develop an annotator model $f_\theta : x \to z$ where $z = (z_1, \ldots, z_N)$. Here $z_j$ denotes the confidence of annotator $A_j$ to correctly label the instance $x$. The annotator model is used to select the annotator for a chosen instance $x$. Finally, we have a knowledge base $KB_j : \mathcal{X} \to \mathcal{Y}$ for each annotator $A_j$ which is populated via already queried instances and is used to infer the label of a new instance $x$ without querying the annotator and thus not consuming budget.

Our method comprises 2 phases - the Boot Phase and the Active Learning Phase. During the Boot Phase, we initiate the bootstrapping process for both the model classifier and the annotator model. Additionally, we initialize the knowledge bases of each annotator. In the Active Learning phase, instances and annotators are chosen iteratively based on the classifier, and annotator models and labels for these instances are obtained through either querying the annotator or through the knowledge base. The instances along with their corresponding labels are then used to further improve the models and the knowledge bases. Figure 1 illustrates the boot phase and the annotator-instance selection process for each active cycle. In subsequent sections we describe in detail each of the components of our framework.

### 3.3 BOOT PHASE

We assume to possess only unlabeled data $U$ in the beginning along with an untrained annotator model $f_{\boldsymbol{\theta}}$, an untrained classification model $g_{\boldsymbol{\phi}}$ and empty knowlege bases for each annotator. Thus, to address the challenge of cold-start, we propose a boot phase which is used to bootstrap both the models and the individual knowledge bases.

We begin by sampling a limited collection of randomly chosen samples from the unlabeled pool of instances $U$. We refer to this as **boot set** $U_B$. The rest of the instances are kept in **active learning set** $U_A$. We proceed by querying all available annotators for each instance in $U_B$ and thus obtaining a complete annotation set $\boldsymbol{l}_i = (l_{i1}, \ldots, l_{iN})$ for every $\boldsymbol{x}_i \in U_B$. Next, we use the boot set instances along with the obtained annotations to train the annotator model $f_{\boldsymbol{\theta}}$. Details about annotator model training is present in section 3.4. Next, for each instance $\boldsymbol{x}_i \in U_B$, we determine the corresponding label $y_i$ using the weighted majority of the annotations $\boldsymbol{l}_i$ where weights are obtained from the annotator model $f_{\boldsymbol{\theta}}$. $\boldsymbol{D}_g = \{(\boldsymbol{x}_i, y_i)\}$ is then used to train the classifier model $g_{\boldsymbol{\phi}}$.

Finally, we use the instances $\boldsymbol{x}_i \in U_b$ along with the corresponding annotation $\boldsymbol{l}_i$ and weights from the annotator model $\boldsymbol{z}_i$ to initialize the knowledge base of each annotator. Complete details about knowledge base creation can be found in 3.5.

### 3.4 ANNOTATOR MODEL

Recent multi-annotator active learning algorithms (**DAAL** Baumler et al. (2023)) randomly choose the annotator to annotate a given instance. This leads to sub-optimal performance since different annotators have different areas of expertise and therefore are not equally likely to label any given instance correctly. Hence, in order to increase the performance of our active learning framework, we choose the most competent annotator for a given instance $\boldsymbol{x}$ by using the annotator model $f_{\boldsymbol{\theta}}$ which predicts the confidence of each annotator to label an instance correctly. The model takes as input an instance $\boldsymbol{x}$ and outputs the confidence score vector $\boldsymbol{z} = (z_1, \ldots, z_N)^T$ where $z_j$ indicates the confidence of annotator $A_j$ to correctly label the instance $\boldsymbol{x}$. The confidence scores are used to determine which annotator is most likely to label the instance correctly.

One major challenge in training such an annotator model is the lack of labeled data. Since we assume that we do not have access to any labeled data, hence it is difficult to train the annotator model. To generate labeled data for training the annotator model, we use annotator agreement to determine the confidence of an annotator. Let $\boldsymbol{l} = (l_1, \ldots l_k)$ be the set of annotations obtained for an instance $\boldsymbol{x}$ and let $l_{maj}$ be the majority label from the set $\boldsymbol{l}$. We assign confidence score $c_j$ to annotator $A_j$ as follows:

$$c_j = \begin{cases} 1 & \text{if } l_j = l_{maj}, \\ 0 & \text{otherwise} \end{cases} \tag{1}$$

In other words, we assign the confidence score of 1 if the label obtained from the annotator matches the majority label else 0. Using equation 1 we generate the confidence scores of all instances which have been queried atleast 3 times (We add this restriction since majority label does not make sense with 2 or less annotations). With this we create the training dataset for the annotator model $\boldsymbol{D}_f = \{(\boldsymbol{x}_i, \boldsymbol{c}_i, \boldsymbol{m}_i)\}$ where $\boldsymbol{x}_i$ has been queried atleast 3 times. Here $\boldsymbol{m}_i = (m_{i1}, \ldots, m_{iN})^T$ is the query mask for the instance $\boldsymbol{x}_i$ where $m_{ij} = 1$ if annotator $A_j$ is queried for instance $\boldsymbol{x}_i$ else $m_{ij} = 0$. To train the model we use masked mean squared error loss defined as follows:

$$L_{\boldsymbol{\theta}} = \frac{1}{|\boldsymbol{D}_f|} \sum_{(\boldsymbol{x}_i, \boldsymbol{c}_i, \boldsymbol{m}_i) \in \boldsymbol{D}_f} \sum_{j=1}^{N} m_{ij}(c_{ij} - z_{ij})^2 \tag{2}$$

### 3.5 KNOWLEDGE BASE

One of the limitations of existing active learning frameworks is that to obtain a label for an instance, they always need to query the annotator. Thus the number of labeled instances that can be used to train the classifier model is limited by the budget $B$. However, one can use the knowledge from the labeled instance queried so far to determine the label of a given instance without querying the annotator. Thus, as an alternative source of labeling without consuming budget, we propose the

creation of knowledge base $KB_j : \mathcal{X} \to \mathcal{Y}$ for each annotator $A_j$. $KB_j$ consists of a set of instances whose labels were already obtained from querying annotator $A_j : K_j = \{(x_{jp}, l_{jp})\}$.

We only keep those instances in the knowledge base of annotator $A_j$ for which the annotator is highly confident. To do so, we add a queried instance $x$ along with its corresponding queried label $l$ to $K_j$ iff $j = \arg\max z$ and $z_j > \eta_a$ where $z = f_\theta(x)$ is the confidence score vector of x obtained from the annotator model and $\eta_a$ is the predefined confidence threshold hyperparameter. Thus we quantitatively capture the expertise of each annotator through their knowledge base $KB_j$.

Our intention through this work is to provide our classification model with a higher number of quality training samples without violating the pre-defined budget. In order to do so we realize the necessity of finding an alternative labelling source except for querying the annotators (which consumes our budget). To obtain free additional labels we intend to quantitatively capture the expertise of each annotator. Therefore we propose to create individual knowledge base for each annotator which shall store the instances along with their predicted labels on which the particular annotator is highly confident.

$KB_j(x)$ denotes the label obtained using the knowledge base $KB_j$ for an instance $x$. $KB_j(x)$ is defined by the following equation:

$$KB_j(x) = \begin{cases} l_{jp} & \text{if } p = \underset{k|(x_{jk}, l_{jk}) \in K_j}{\arg\max} \ sim\,(x, x_{jk}) \text{ and } sim\,(x, x_{jp}) > \eta_{sim} \\ -1 & \text{otherwise} \end{cases} \tag{3}$$

i.e we first select the instance $x_{jp}$ from the knowledge base which is most similar to the new instance $x$ and assign it the label $l_{jp}$ which was given to $x_{jp}$ by annotator $A_j$ if similarity between $x$ and $x_{jp}$ is greater than some similarity threshold $\eta_{sim}$ otherwise we do not assign any label (indicated by -1). In this way we obtain label for a new instance $x$ without querying the annotator $A_j$ based on the instances the annotator had labeled before.

## 3.6 ACTIVE PHASE

After having bootstrapped the classifier model $g_\phi$, the annotator model $f_\theta$ and the knowledge bases $\{KB_j\}_{j=1}^N$, we begin our active learning phase. First, we calculate remaining budget $B_{active} = B - |U_B| \times N$, by subtracting the budget that we utilized in the boot phase. Next, we iteratively select an instance in the active learning set $U_A$ and a corresponding annotator to query based on our instance and annotator selection strategies. After obtaining the label from the annotator, we suitably update our classifier training dataset $D_g$ and our annotator training dataset $D_f$ and train both the models. We also update the knowledge base of the queried annotator with the new instance-label pair as mentioned in section 3.5. Since model training is expensive, we train the models after a batch of iterations and not after every single iteration. This process continues till the budget $B_{active}$ is exhausted. We now describe in details our instance selection and annotator selection strategies in the subsequent sections.

### 3.6.1 INSTANCE SELECTION

Traditional active learning methods use uncertainty based techniques such as entropy to select the instances that are needed to be annotated. While such methods perform reasonably well in the case of single annotator where the obtained labels are accurate, their effectiveness reduces in the presence of multiple noisy annotators. This is because due to the presence of noisy labels, the classifier model might become incorrectly confident about certain instances and not query them again. This will lead to decrease in the performance of the classifier model. To tackle this issue, we propose a novel instance selection technique **AAD** which utilizes **A**nnotator **A**greement-**D**isagreement to determine how noisy the label is for a particular instance.

To begin with, we define two sets of instances: the explore set $E$ (which will contain instances that have never been queried) and the queried set $Q$ (which will contain the instances that have been queried atleast once). $E$ is initialized with all the instances in the active pool $U_A$ and $Q$ is initially empty. We next define an exploration parameter $\epsilon$. In every active learning iteration, with probability $\epsilon$ we choose to explore a new instance from $E$ and with probability $1 - \epsilon$ we choose to relabel a

previously annotated instance from $Q$. Whenever we decide to explore, we use the entropy measure to choose the instance which the model is least confident about. Since the instances in the explore set have never been queried before, they do not suffer from the issue of noisy labels and hence selecting instances based on entropy becomes a viable approach. Once an annotator has been queried for the selected instance, it is then removed from the explore set and added to the queried set.

On the other hand, whenever we decide to relabel an instance from the queried set, we first calculate annotator disagreement score $ad\_score$ for each instance in the queried set and choose the instance with the highest disagreement score for relabeling. To calculate the $ad\_score$ for an instance $x_i$, we first calculate the aggregate confident scores of each class label $Y$ based on the annotators who have previously annotated $x_i$ as $Y$. Next we obtain the weighted majority label $Y_{maj}$ based on the aggregated confidence scores. $ad\_score$ is defined as the 1 minus the average of the absolute difference between the confidence score of each class label $Y$ with $Y_{maj}$ which is mathematically expressed as follows:

$$ad\_score = 1 - \frac{1}{C-1} \sum_{Y \neq Y_{maj}} |conf\_score(Y_{maj}) - conf\_score(Y))| \qquad (4)$$

where $conf\_score(Y) = \sum_j z_{ij} \times m_{ij}$. Here $z_{ij}$ is the $j^{th}$ element of the confidence vector $z_i = f_\theta(x_i)$ and $m_{ij}$ is the $j^{th}$ entry of the query mask $m_i$. The intuition behind the annotator disagreement score is that the score should be higher when equally confident annotators disagree with each other and the score should be lower when a more confident annotator disagrees with a less confident annotator with the score being 0 when all annotators agree with full confidence (confidence score of 1). Once $ad\_score$ is calculated for all the instances in $Q$, the instance with the highest disagreement score is chosen for relabeling. Once all the annotators of a chosen instance are queried, it is then removed from the queried set $Q$ as well.

### 3.6.2 ANNOTATOR SELECTION

Once an instance $x_i$ is selected for annotation, we decide which annotator to query using the annotator model $f_\theta$. We select the annotator $A_j$ who has the highest confidence score for the instance $x_i$ amongst the annotators remaining to be queried for that instance. given by the equation: $j = \arg\max_k (z_{ik} \times (1 - m_{ik}))$ where $z_{ik}$ is the $k^{th}$ entry of the confidence vector $z_i = f_\theta(x_i)$ and $m_{ik}$ is the $k^{th}$ entry of the query mask $m_i$.

Next we try to fetch the label from the knowledge base of annotator $A_j$ (equation 3). If a label is obtained, we annotate the instance $x_i$ with the label $KB_j(x)$ without consuming any budget. If a label could not be fetched from the knowledge base, then we actually query the annotator to get the label. We then add the instance along with the queried label to the annotator's knowledge base if it satisfies the condition mentioned in 3.5. Finally, we update the classifier training data $D_g$ and the annotator training data $D_f$ with the new instance-label pair. We train the models using the updated datasets after a batch of iterations have passed.

## 4 EXPERIMENT

In this section, we introduce the datasets we use, experimental details, and the baseline methods against which we compared our approach.

### 4.1 DATASETS

We selected 11 diverse sets of publicly available UCI benchmark data sets Lichman et al. (2013) from various domains to ensure the robustness and generalizability of our framework. We also selected two real-world text classification datasets such as Reports Mozilla and Reports Compendium Hernández-González et al. (2018), these were annotated by real-world annotators, providing an authentic and practical dimension to our evaluations. These datasets exhibit differences in features, classes, annotators, and annotator accuracies. For the remaining datasets, annotators were simulated based on instance-dependent performance Fang et al. (2012). In this simulation, the probability of

| Characteristics | min | max | average |
|---|---|---|---|
| features | 6 | 100 | 36 |
| number of classes | 2 | 8 | 3 |
| number of annotators | 4 | 6 | 4 |
| annotator accuracies | 34 | 84 | 53 |
| size of unlabelled samples | 124 | 577 | 350 |
| size of test samples | 84 | 384 | 233 |

Table 1: Dataset Statistics

an annotator providing a correct annotation depended on both the annotator's characteristics and the cluster to which the instance belonged. To determine these clusters, we employed a k-means clustering approach Hartigan & Wong (1979) with $k$ equal to the number of annotators. Basic data statistics have been summarized in Table 4.1

## 4.2 Setup and Baselines

We maintained a consistent 60-40 train-test split for all datasets throughout our experiments. In the CLA-RA framework, the boot phase utilized 3% of the training data making sure that the number of boot instances is at least equal to the number of classes. We set the annotation budget at 20% of the total available annotations for each dataset, maintaining this constant across all methods. No such ground-truth labels are available for the training set. To ensure fairness in the comparison, considering CLA-RA's boot and active learning phases, we deducted the total annotations used during the boot phase from the budget, establishing a uniform total number of annotations for all methods. We set $0.75$ as exploration threshold $\epsilon$, $0.7$ as confidence threshold $\eta_a$ and $0.99$ as similarity threshold $\eta_{sim}$ for considering the KB. We used Extra_trees Geurts et al. (2006) as the classifier model $g_\phi$ and a simple 2 layer feed-forward neural network with layer norm and ReLU Agarap (2018) as the activation function. Dimension of both the hidden layers were chosen to be 32.

We compared the proposed algorithm against the best state-of-the-art methods: Cost Effective Active Learning (CEAL) proposed by Huang et al. (2017), Proactive proposed by Moon & Carbonell (2014), ALIO proposed by Chakraborty (2020) and we implemented Disagreement aware Active Learning (DAAL) proposed by Baumler et al. (2023).

## 5 Results

In this section, our primary focus centers on evaluating the test accuracy of the classifier trained using data for which labels were acquired during the active learning phase. Our experimental design aims to provide comprehensive insights by addressing the following key research questions:

- **RQ1:** Does CLA-RA framework effectively leverage the collective wisdom of multiple annotators to improve classifier performance compared to existing active learning methods?
- **RQ2:** How effective is the instance-dependent annotator model in helping to choose the best annotator for a given instance?
- **RQ3:** Does the strategic instance selection mechanism improve the efficiency of the annotation process?
- **RQ4:** How accurate is the annotator knowledge base in capturing the proficiency and how helpful it is in the overall active learning framework

## 5.1 Effectiveness of CLA-RA Framework in Harnessing Collective Annotator Wisdom

We compare the accuracy of the classifier of CLA-RA framework with state-of-the-art multi-annotator active learning algorithms with the same budget and train test split over 13 datasets comprising both binary and multi-class setups. Table 2 provides an overview of the performance of various active learning methods on these datasets. Notably, our proposed CLA-RA framework outperforms the other approaches in 10 out of the 13 datasets. The CEAL method demonstrates competitive performance in the titao and mushroom's datasets. However, its performance is not consistently

| Dataset | CEAL | ALIO | Proactive | DAAL | CLA-RA |
|---|---|---|---|---|---|
| car | 59.17 | 67.92 | 55.83 | 62.08 | **73.33** |
| ecoli | 69.63 | 74.81 | 74.07 | 75.55 | **83.70** |
| ionosphere | 70.92 | 68.79 | 70.92 | 75.88 | **92.08** |
| mushroom | **92.50** | 91.67 | 88.33 | 84.16 | 90.83 |
| reports-compendium | 50.13 | 50.91 | 41.04 | 42.33 | **52.21** |
| reports-mozilla | 53.33 | 48.52 | 54.81 | 59.25 | **60.26** |
| ringnorm | 66.25 | 68.33 | 65.83 | 67.08 | **92.08** |
| sonar | 70.24 | 71.43 | 59.52 | 70.23 | **73.81** |
| spambase | 76.67 | 76.25 | 74.58 | 79.16 | **82.92** |
| splice | 48.75 | 50.83 | 47.50 | 40.83 | **62.08** |
| titato | **73.33** | 71.67 | 70.83 | 57.5 | 71.25 |
| vehicle | 59.59 | 61.65 | **64.31** | 50.14 | 62.54 |
| krvsp | 68.33 | 70.0 | 68.33 | 60.41 | **78.75** |

Table 2: Accuracy (in percentage) of the various Active Learning approaches. Best results are shown in bold.

high across all datasets. This inconsistency in performance is also observed in the *Proactive* across various datasets. When considering the average classifier accuracy across all datasets, the performance ranking of the various models can be summarized as follows: CLA-RA consistently achieves the highest accuracy, followed by ALIO, CEAL, *Proactive*, and DAAL.

In the subsequent sections, we'll analyze our model's components and their role in achieving top accuracies across datasets. To improve readability, we've experimented on a curated subset of 8 datasets, balancing label types, annotator skills, and feature spaces.

## 5.2 QUALITY OF LABELS ACQUIRED VIA INSTANCE-DEPENDENT ANNOTATOR

Next, we assess the impact of the Annotator Model $f_\theta$ on annotator selection and label quality, addressing the second research question. In this experimental analysis, we examined the accuracy of labels provided by the annotators across all methods. Figure 2 presents the percentage accuracy of labeling unlabeled instances for the subset datasets by each method. Notably, our proposed approach attains the highest labeling accuracy in all 8 datasets. Thus using $f_\theta$ annotator model helps in selecting the most suitable annotators for annotating each sample.

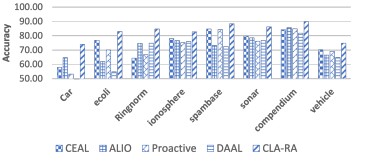 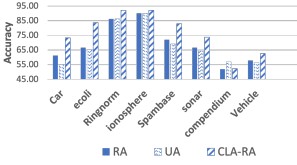 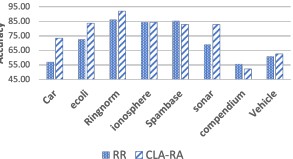

Figure 2: Accuracy of acquired labels

Figure 3: Instance selection strategy

Figure 4: Accuracy of Classifier on Test data

## 5.3 EVALUATING THE AAD INSTANCE SELECTION STRATEGY

We delve deeper into the crucial role played by our AAD instance selection strategy, addressing the third research question. We compare CLA-RA's performance against two alternatives: (i) Random sampling (RA), where samples are randomly selected and labeled using the $f_\theta$ annotator model, and (ii) Uncertainty sampling (UA), which targets unlabeled samples with the highest classification uncertainties for annotation using the proposed $f_\theta$ annotator model.

Figure 3 illustrates the classifier accuracy achieved with these various instance selection mechanisms. Remarkably, our proposed model outperforms the alternatives in 7 out of 8 datasets. This performance boost can be attributed to the instance selection strategy, as it constitutes the variable component among the three approaches. The balance between exploration and exploitation, achieved by selecting instances based on the level of disagreement, contributes significantly to reducing noise in the training data, ultimately enhancing the performance of the classifier model.

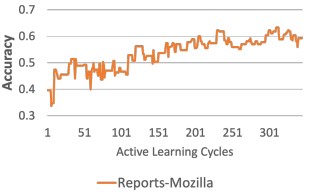 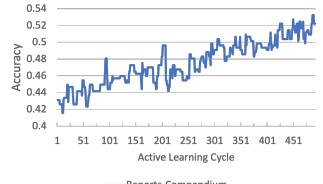 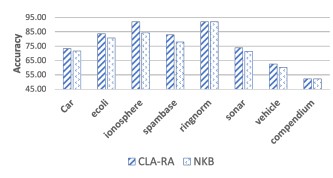

Figure 5: Test Accuracy after each active learning iteration

Figure 6: Test Accuracy after each active learning iteration

Figure 7: Accuracy with and without using Knowledge base

To gain deeper insights into the effectiveness of both the instance-dependent annotator model and the strategic instance selection model, We compared the performance of CLA-RA, with a random annotator-instance selection model, referred to as RR. Figure 4 displays the results. Our observations indicate that CLA-RA consistently outperforms RR by around 11-16 percentage points in datasets like car, ecoli, and sonar. Notably, the proposed approach proves to be particularly valuable in scenarios where the average annotator accuracy is relatively low, resulting in a higher probability of errors in the acquired labels. Conversely, in scenarios with less error-prone annotators, RR can still achieve reasonable results. This underscores the adaptability of our framework to varying levels of annotator expertise and its ability to enhance classifier accuracy, particularly in challenging annotation scenarios.

We present the classifier's accuracy for two datasets, reports-mozilla and reports-compendium, annotated by human experts. Figures 6 and 5 illustrate this. The x-axis represents active learning cycles, and the y-axis denotes test set accuracy. Notably, as cycles progress, the model's performance consistently improves. This improvement is attributed to the model's ability to effectively select examples for labeling and re-labeling, all orchestrated by the selection of the most suitable annotator for each instance.

### 5.4 GAUGING THE UTILITY OF ANNOTATOR EXPERTISE INFERENCE MODEL

To assess the utility of our Annotator Expertise Inference model, we compare the performance of CLA-RA with an approach where expertise inference model is not used to acquire labels for new instances, we refer this approach as NKB. Figure 7 shows the accuracy of the classifier for both approaches. Notably, when the expertise inference model was not employed, there was a substantial decrease of 2-5% in the classifier's performance across the datasets. This decline can be attributed to the model's ability to acquire additional high-quality labeled data without explicitly querying the annotator, effectively optimizing the utilization of the annotation budget.

## 6 CONCLUSION

In this paper, we proposed CLA-RA, a novel framework for enhancing the acquisition of high-quality labeled data in the presence of multiple annotators. By harnessing the collective expertise of annotators and strategically selecting the most suitable annotator-instance pairs, our approach consistently outperforms state-of-the-art methods across diverse datasets. Our framework's notable advantage lies in its ability to optimize the utilization of the annotation budget. It strikes a delicate balance between exploration and exploitation, effectively reducing noise in the training data. This equilibrium not only enhances classifier accuracy but also ensures the efficient allocation of resources during the annotation process. By minimizing unnecessary labeling efforts and concentrating on the most informative instances, our method also maximizes the efficiency of the annotation process.

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
