# OpenReview forum: "CLA-RA: COLLABORATIVE ACTIVE LEARNING AMIDST RELABELING AMBIGUITY"
_ICLR.cc/2024/Conference — Submitted to ICLR 2024_

### Official Review · Reviewer_8mJ8 · 2023-10-28

**Soundness:** 4 excellent
**Presentation:** 4 excellent
**Contribution:** 2 fair
**Rating:** 3
**Confidence:** 5

**Summary:**

This paper introduces a new scheme for active learning based (re-)labeling of ML datasets that leads to improved classification accuracy of the classifiers resulting from training on the (re-)labeled datasets. The key idea of the paper is to combine crowd-sourced annotation from annotators who are not necessarily experts and thus potentially unreliable (at least to some extent) with confidence estimations for individual annotators — combined these two ideas evidently lead to substantially improved classifiers, as evaluated on a range of standard UCI ML datasets.

**Strengths:**

This is a very solid paper that makes an important contribution to the active learning community. I applaud the authors for a clear, technically sound presentation of a clever idea that evidently (as shown by the experimental evaluation) leads to improvements in classification accuracy for classifiers that are trained through active learning. While the related work part is a bit short, it covers relevant concepts and previous work in the field. The technical approach is nice: combining crowdsourcing with active learning, thereby taking into account that crowd annotators are not necessarily experts and then turning this around by assigning assumed confidence values to individual annotators through the majority voting on multiply labels samples and looking at consistency amongst individual annotators and their performance (with regard to the majority votes) is clever. Combining this with the budget constraint that is often used in active learning scenarios evidently leads to an effective active learning scheme. The experimental evaluation is solid. The authors did a good job in experimenting over a range of datasets (even though some more details on these should have been given, e.g., in an appendix, to make the paper more comprehensive). As usual in the field, alas, no statistical significance analysis has been performed — yet the numerical gains are large enough such that this reviewer (who has experience with those datasets) believes that the gains actually are significant (see weaknesses and questions). The presentation of the approach is solid and the paper is easy to follow.

**Weaknesses:**

There are a few issues with the paper that should be considered: Extend the related work a bit (there should be space); add an actual statistical significance analysis; justify why accuracy is used as evaluation measure (and not macro F1 as it would be more appropriate for at least some of the rather imbalanced datasets; add an appendix where more details are given, e.g., on the datasets and on the classification pipeline.

My biggest concerns with this paper, however, are scope and potential impact. ICLR is, as per its own definition, a venue that focuses on representation learning / deep learning. This paper is essentially a core machine learning paper, yet it does not actually cover representation learning nor deep learning. As such, I see this paper in its current form slightly out of scope for ICLR. The second concern (impact) comes from the fact that active learning has been around for quite a while (even including the crowd sourcing component as identified by the authors), and the field is somewhat saturated. The presented approach is clever and effective for a range of standard classification tasks but I wonder whether this would actually be of substantial enough interest for the ICLR audience. If the authors would link their work closer to recent applications / problems in the deep learning / representation learning field it probably would gain more attention / impact.

**Questions:**

See my comments above with suggestions / questions that could be addressed in an appendix etc.

---

> ### Author Response · Authors · 2023-11-21
>
> **Q: accuracy is used as evaluation measure**
>
> Accuracy is chosen as the metric since most of the SOTA baselines we have compared with have
> used accuracy as their metric for same / similar datasets.

---

### Official Review · Reviewer_T5Kg · 2023-10-30

**Soundness:** 3 good
**Presentation:** 2 fair
**Contribution:** 2 fair
**Rating:** 3
**Confidence:** 5

**Summary:**

This paper proposes a framework CLA-RA to solve the problems of active learning strategy in crowdsourcing. The proposed method consists of an instance selection approach, an annotator confidence model, and an inference model. The proposed method compares with several baselines on 13 datasets. A comprehensive insight into each component is also applied.

**Strengths:**

1. The proposed method is technically sound.
2. The framework is clear and easy to follow.
3. The comprehensive insights for each component of the proposed method are interesting.

**Weaknesses:**

1. Notations are not defined clearly. Also, the usage of notations is really confusing. This is the most important thing that hinders understanding. Here are some examples. The notations for features of an instance are very similar to the notation of the instance itself. And, the last $x$ in the first paragraph of Section 3.1 is misused; The condition in the second paragraph of Section 3.5 is ambiguous. An indicator (j) cannot be equal to a vector (z).
2. The selection of the datasets is not very reasonable. The characteristics of all datasets are not clear. This is my main concern about the effectiveness of the experiment. Detailed traits of each dataset should be exhibited. The number of annotators is too small. This condition is really beneficial to the proposed method which drops the convincing of the experiment.
3. Hyper-parameters are too much. The experiment part lacks an explanation for the specific values of these hyper-parameters. Due to the core component being the annotator model, a well-work $ f_{\theta} $ is really important to the proposed method. However, there is no mechanism to guarantee the training result of that $ f_{\theta} $ is ‘good’.

**Questions:**

1. How were the hyper-parameters, i.e. ratio of training data, annotation budget, exploration threshold, confidence threshold, and similarity threshold, selected in the experiment?
2. How about the time complexity of the proposed method compared with other methods? When the number of workers becomes greater, due to the training phase, the proposed method will
3. How to guarantee that the annotator model works well? If the training data is unbalanced or responses from annotators have great bias, the annotator model will fail. Also, too small a quantity of training datasets cannot overcome the random prediction brought by the random initialization of the neural network.

---

> ### Author Response · Authors · 2023-11-21
>
> **The condition in the second paragraph of Section 3.5 is ambiguous. An indicator (j) cannot be equal to a vector (z).**
>
> The condition is **j = argmax of the vector z** which is basically the index of the maximum element in z.
>
> **Query about the datasets.**
>
> The datasets chosen here are a combination of datasets present in previous works in these domain (eg CEAL, ALIO, Proactive etc) which are simulated from standard UCI datasets. We have provided a summary of the datasets in table 1 due to lack of space. However we agree to more details about the individual datasets in the appendix.
>
> **Q: How were the hyper-parameters, i.e. ratio of training data, annotation budget, exploration threshold, confidence threshold, and similarity threshold, selected in the experiment?**
>
> Determining the correct set of hyper-parameter values is a challenging task here, since we assumed
> no labelled validation set to be available. We used some heuristics to decide on the hyperpaameters. For
> example, learning rate was decided based on looking at the convergence of the training loss,ratio of boot
> to active learning samples were decided based on the how many examples are needed to have atleast
> 1 example for each class, confidence and similarity threshold were decided based on looking at the
> distribution of the respective values. We experimented with various similarity thresholds ranging from 0.7 to 0.95 and we observed that for most of the datasets there was not much change in the performance of the classifier with varyingsimilarity threshold. This indicates that some datasets are well separated in the embedding space However we agree that these are very crude ways to determine the
> hyperparameters, therefore we decided to fix the value of the hyperparameters for all the datasets to
> show the generalizability of these values. However if the constraint of not having any validation dataset is
> relaxed, one can tune the hyperparameters based on the validation dataset to possibly improve the
> numbers even further.
>
> **Q: How about the time complexity of the proposed method compared with other methods?**
>
> Time complexity wrt the no. of annotators (N) is linear because operations in KB is linear in N and the
> change in the annotator model is also in the output layer which is linear in N.
>
> **Q: How to guarantee that the annotator model works well? If the training data is unbalanced or responses from annotators have great bias, the annotator model will fail.**
>
> We showcase the efficacy of our annotator model empirically in figure 2 where we show the accuracy of the labels obtained from the annotator model. Many of the datasets , especially the multi class datasets such as car, ecoli etc have unbalanced distribution of labels for which our model outperforms the baseline by a large margin. However as mentioned in the reply of review 1, in the situation where most of the time, majority is incorrect, our approach will not perform optimally, which remains a drawback of not only this approach but the other baselines as well.
>
>  **too small a quantity of training datasets cannot overcome the random prediction brought by the random initialization of the neural network**
>
> This is the reason of choosing to relabel an instance with some probability at each step. For instances on which the Annotator model is not very confident, those instances will be chosen during the relabelling step and the annotator model will be trained for those instance with more label information.

---

### Official Review · Reviewer_hpUt · 2023-11-02

**Soundness:** 2 fair
**Presentation:** 3 good
**Contribution:** 2 fair
**Rating:** 5
**Confidence:** 4

**Summary:**

The paper introduces the CLA-RA framework, a novel approach designed for optimized data labeling in machine learning, specifically addressing noise and inaccuracies from crowd labelers. Key innovations of CLA-RA include an 'Annotator Disagreement-Based Instance Selection Mechanism' for better noise detection, an 'Instance-Dependent Annotator Confidence Model' to streamline annotation, and the 'Annotator Inference Method' to build annotator-specific knowledge repositories based on consensus. Empirical tests across 13 datasets indicate a marked improvement in classifier accuracy with CLA-RA, underscoring its potential to enhance data labeling efficiency and quality in active machine learning.

**Strengths:**

To address the issue of active learning with multiple annotators, the paper proposes a suite of intricate mechanisms, including a knowledge repository, instance-related selection mechanisms, and annotator selection strategies, which are highly advantageous for the implementation of crowdsourced learning in practical settings.

**Weaknesses:**

The experimental section of the article appears overly concise, with the most critical deficiency being the absence of ablation studies for relevant components, which leaves unclear which step plays a pivotal role. Additionally, the section lacks essential parameter analyses, such as those for the exploration threshold \epsilon, the confidence threshold \eta, and others...

**Questions:**

Q1: The authors claim in the experimental section that Figures 6 and 5 demonstrate the efficiency of their example selection strategy for labeling and re-labeling; however, no comparative analysis is presented within these figures. In my opinion, a comparison with alternative selection strategies, such as random annotator selection, should be included. The efficacy of the sample selection strategy would be more convincingly demonstrated if the same level of accuracy could be achieved with fewer queries.

Q2: In the method proposed by the article, a weighted majority scheme is employed to compute the confidence among different annotators. Numerous extant methods exist to calculate confidence, such as utilizing the L1 norm of predictions. My question concerns the decision to employ the weighted majority algorithm—a relatively conventional approach—over more recent methodologies. What was the rationale behind this choice?

---

> ### Author Response · Authors · 2023-11-21
>
> **Q: My question concerns the decision to employ the weighted majority algorithm—a relatively conventional
> approach—over more recent methodologies. What was the rationale behind this choice?**
>
> We used weighted majority as it was performing the best across all the datasets amongst few other
> techniques. However we do agree with the reviewer’s comment that there might be better techniques to
> infer annotator confidence.

---

### Official Review · Reviewer_L6GL · 2023-11-06

**Soundness:** 1 poor
**Presentation:** 2 fair
**Contribution:** 2 fair
**Rating:** 3
**Confidence:** 4

**Summary:**

The authors present the CLA-RA framework, an active learning approach to selecting instances to be labeled and the annotators who will label them in a crowd-source setting. The goal of this framework is to best use a annotation budget to achieve the highest accuracy for a classifier. To achieve this the framework has two main components, and instance selection process and an annotator selection process.  The authors evaluate over 13 datasets (11 of which have simulated multiple annotator labels, and two that have legitimate multiple ratings). The authors show how their active learning framework out performs other similar approaches across many of the datasets (10 out of the 13).

**Strengths:**

The approach taken by the authors is sensible, splitting the components of selecting an instance to be labeled and then selecting whom should do the labeling. The paper provides thorough citations.

**Weaknesses:**

I have concerns about the results. There are thirteen datasets, but often the authors only show eight datasets when presenting their results. It also isn't clear if the improvements are statistically significant. The authors have no description of how they simulated the multiple ratings, the make passing reference to another paper. Given the importance of this detail the authors should explain how they simulated the ratings AND reference the paper. Most of the figures have a Y-axis that DOES NOT start at 0, thus inflating their improvements. In the two non-simulated datasets the proposed framework barely surpasses past approaches. For figure five and seven the authors show no baseline. How would random selection of instances and annotators performed in this case? They hand-wave that their approach works best when raters are more error prone.

**Questions:**

- Why does the approach not work as well as CEAL for Mushroom and Vehicle?
- Would this approach work in high-dimension domains, thus making the similarity function more of an issue?
- The Annotator Model makes a big assumption that the majority will be correct, can this work in cases where the majority isn't correct?

---

> ### Author Response · Authors · 2023-11-21
>
> **There are thirteen datasets, but often the authors only show eight datasets when presenting their results.**
>
> We decided to show eight datasets for the ablation study results to maintain the clarity of the charts and not make them too crowded.
>
> **Q: Why does the approach not work as well as CEAL for Mushroom and Vehicle?**
>
> CEAL actually uses a small set of labeled data to initialise their algorithm. We went ahead with a much difficult setting of having no labeled data to bootstrap. Even without any labeled data, we outperform CEAL in most datasets.
>
> **Q: Would this approach work in high-dimension domains, thus making the similarity function more of an issue**
>
> For higher dimensional cases such text and images, one can use pretrained models to obtain the
> embeddings. For example for text, one might use models such as BERT to encode the text into an
> embedding vector and for images, one can use CNNs to get the embeddings. These embeddings can
> then be passed to the similarity function
>
> **The Annotator Model makes a big assumption that the majority will be correct, can this work in cases where the majority isn't correct?**
>
> The goal of having an annotator model which predicts annotator confidence taking in instance features as input was to help improve the labelling of an instance even when majority is not correct for that particular instance. Having said that, the way the annotator model is trained, if for most of instances, majority is incorrect, then the annotator model will be incorrectly trained and will not give optimal results. We agree that this is a drawback of the method, but all the existing methods which do not use any kind of prior information about the labellers (including having few labeled examples) also suffer from this drawback.

---

### Meta-Review · Area_Chair_jYwQ · 2023-12-03

**Metareview:**

This paper proposes an active learning approach for crowdsourcing, focusing on determining which instances should be labeled and which annotators should provide these labels, with the aim of optimizing classifier accuracy. The authors evaluated their method across 13 datasets, demonstrating that their approach outperforms other baselines.

Overall, all reviewers agree that the paper addresses a practical question, and the proposed approach is sensible. However, they have raised various concerns, leading them to believe that the paper in its current form does not yet meet the acceptance bar yet. The primary concerns revolve around the evaluations, including the authors presenting only a subset of results, the absence of statistical tests to assess result significance, the lack of ablation studies to illustrate the impact of different components, and the choice of evaluation metric, among others. We hope the authors will find these comments from the reviewers helpful in refining the next version of the manuscript.

**Justification For Why Not Higher Score:**

There is a consensus that the current paper is not up to the bar for publication at ICLR.

**Justification For Why Not Lower Score:**

N/A

---

### Decision · Program_Chairs · 2024-01-16

Reject